# OXR1 maintains the retromer to delay brain aging under dietary restriction

Kenneth A. Wilson [1,2], Sudipta Bar [1], Eric B. Dammer [3], Enrique M. Carrera[1], Brian A. Hodge[1], Tyler A. U. Hilsabeck [1,2], Joanna Bons [1], George W. Brownridge III[1], Jennifer N. Beck[1], Jacob Rose[1], Melia Granath-Panelo[1], Christopher S. Nelson[1], Grace Qi [1], Akos A. Gerencser [1], Jianfeng Lan[1,4], Alexandra Afenjar[5,6], Geetanjali Chawla [7], Rachel B. Brem[1,2,8], Philippe M. Campeau [9], Hugo J. Bellen [10], Birgit Schilling [1], Nicholas T. Seyfried [3], Lisa M. Ellerby [1,2] ✉ & Pankaj Kapahi [1,2] ✉

Dietary restriction (DR) delays aging, but the mechanism remains unclear. We identified polymorphisms in *mtd*, the fly homolog of *OXR1*, which influenced lifespan and *mtd* expression in response to DR. Knockdown in adulthood inhibited DR-mediated lifespan extension in female flies. We found that *mtd*/*OXR1* expression declines with age and it interacts with the retromer, which regulates trafficking of proteins and lipids. Loss of *mtd*/*OXR1* destabilized the retromer, causing improper protein trafficking and endolysosomal defects. Overexpression of retromer genes or pharmacological restabilization with R55 rescued lifespan and neurodegeneration in *mtd*-deficient flies and endolysosomal defects in fibroblasts from patients with lethal loss-of-function of *OXR1* variants. Multi-omic analyses in flies and humans showed that decreased Mtd/OXR1 is associated with aging and neurological diseases. *mtd*/*OXR1* overexpression rescued age-related visual decline and tauopathy in a fly model. Hence, OXR1 plays a conserved role in preserving retromer function and is critical for neuronal health and longevity.

Aging is a leading contributor to cognitive decline, but dietary restriction (DR) delays aging across species and slows the progression of neurodegenerative diseases[1]; however, the mechanisms that mediate the protective effects of DR in the brain are not fully understood[2]. Additionally, factors such as natural genetic variation greatly influence response to DR, leading to the concept of precision nutrigeroscience to understand how differences between individuals and across tissues modulate responses to diet and influence healthspan and lifespan[2]. We previously used the *Drosophila* Genetic Reference Panel (DGRP)[3] to better understand the genetic effectors of DR-based lifespan response[4], and in this current study, we focus on polymorphisms in the

[1]Buck Institute for Research on Aging, Novato, CA 94945, USA. [2]Leonard Davis School of Gerontology, University of Southern California, Los Angeles, CA 90089, USA. [3]Department of Biochemistry, Emory University School of Medicine, Atlanta, GA 30322, USA. [4]Guanxi Key Laboratory of Molecular Medicine in Liver Injury and Repair, The Afilliated Hospital of Guilin Medical University, Guilin 541001 Guanxi, China. [5]Assistance Publique des Hôpitaux de Paris, Unité de Génétique Clinique, Hôpital Armand Trousseau, Groupe Hospitalier Universitaire, Paris 75012, France. [6]Département de Génétique et Embryologie Médicale, CRMR des Malformations et Maladies Congénitales du Cervelet, GRC ConCer-LD, Sorbonne Universités, Hôpital Trousseau, Paris 75012, France. [7]RNA Biology Laboratory, Department of Life Sciences, School of Natural Sciences, Shiv Nadar Institute of Eminence, NH91, Tehsil Dadri, G. B. Nagar 201314 Uttar Pradesh, India. [8]Department of Plant and Microbial Biology, University of California, Berkeley, 111 Koshland Hall, Berkeley, CA 94720, USA. [9]Centre Hospitalier Universitaire Saint-Justine Research Center, CHU Sainte-Justine, Montreal, QC H3T 1J4, Canada. [10]Departments of Molecular and Human Genetics and Neuroscience, Neurological Research Institute, Texas Children's Hospital, Baylor College of Medicine, Houston, TX 77030, USA. ✉e-mail: lellerby@buckinstitute.org; pkapahi@buckinstitute.org

gene *mustard* (*mtd*), which associate with DR-dependent longevity. Genetic variants in the human homolog of *mtd*, *Oxidation Resistance 1 (OXR1)*, are associated with cerebellar atrophy, hypotonia, language delay, and seizures[5], and its overexpression improves survival in a mouse model of amyotrophic lateral sclerosis (ALS)[6]. Despite the protective properties of OXR1, its biochemical mechanism remains unknown. Here, we identify the dietary and genetic factors which regulate *mtd*/*OXR1* and demonstrate its necessity for the maintenance of the retromer complex, which is a heteropentameric complex of proteins necessary for recycling transmembrane proteins and lipids from endosomes to the *trans*-Golgi network or the cell membrane[7,8]. We further show that *mtd*/*OXR1* regulates a network of genes that are essential for protection against brain aging and neurodegenerative diseases across flies and humans.

## Results

### Neuronal expression of mtd is required for lifespan extension upon DR

To identify regulators of the protective effects of DR, we measured lifespan under *ad libitum* (AL, 5% yeast) or DR (0.5% yeast) conditions in female flies from 160 DGRP strains[4]. We identified genetic variants in five genes which significantly associated with extreme longevity upon DR (Supplementary Data 2). Of these genes, only *Ferredoxin* (*Fdxh*) and *mustard* (*mtd*) have human orthologs (*ferredoxin 2* and *Oxidation Resistance 1*, *OXR1*, respectively). We found that an allele in *mtd* consisting of a single-nucleotide polymorphism and an insertion of 101 base pairs associated with reduced lifespan under DR, but not AL, when compared to the wild-type allele (Fig. 1a, Supplementary Fig. 1a, Supplementary Data 2)[3]. In female flies, whole-body *mtd*[RNAi] driven by *Act5c-GS-Gal4* (Fig. 1b), but not *Fdxh*[RNAi] (Supplementary Fig. 1b), resulted in DR-specific reduction in lifespan, suggesting the importance of *mtd* expression in DR-mediated longevity. In contrast, a strong null allele *mtd-T2A-Gal4* strain (*mtd*[MI02920-T2A-Gal4])[5] caused lethality in development in approximately 99% of flies, and DR did not extend lifespan in the flies that survived to adulthood (Fig. 1c). *OXR1* loss in humans is associated with severe neurological defects and premature death[5], whereas overexpression provides protection in a mouse model of amyotrophic lateral sclerosis[6], but its cellular mechanism of action is unknown. *OXR1* is often studied for its role in oxidative stress response[9,10], but Wang et al. showed that its loss causes lysosomal dysfunction independent of oxidative stress[5]. Thus, identifying the mechanism of OXR1 will elucidate a valuable target for neurodegenerative diseases and aging.

DR induced a sevenfold increase in *mtd* mRNA expression in the head (Fig. 1d). Neuronal *mtd*[RNAi] driven by the constitutively active *elav-Gal4* induced developmental defects and severe lifespan reduction under both diets compared to TRiP (empty vector) control (Fig. 1e), similar to whole-body loss of *mtd*. These phenotypes were not observed with intestine- or glia-specific *mtd*[RNAi] (Supplementary Fig. 1c, d). To evaluate the impact of neuronal *mtd*[RNAi] with age, we used the conditional RU486-inducible *elav-GS-Gal4* driver to induce a milder knockdown only in adulthood. This eliminated DR-mediated lifespan extension but did not alter lifespan on AL in females, whereas lifespan was reduced under both diets in males (Fig. 1f and Supplementary Fig. 1e, f). There was no change in fecundity between RNAi and control flies (Supplementary Fig. 1g). The TLDc domain in Mtd/OXR1 is responsible for neuroprotection[5,11], so we overexpressed truncated human *OXR1* containing this domain (*hOXR1*[OE]) in neurons of flies that harbored the *mtd* null allele and observed a complete rescue of lifespan and DR response (Fig. 1g). Neuronal *hOXR1*[OE] in a wild-type background extended lifespan only under DR (Fig. 1h), but also extended lifespan under AL conditions with stronger, constitutively active expression throughout life (Fig. 1i). *mtd* expression was significantly increased across life in wild-type flies under DR, but this expression declined from day 7 to day 21 on both diets (Supplementary

Fig. 1h). Overall, these results demonstrate that neuronal *mtd* expression is elevated by DR, declines with age, and is necessary for DR-mediated lifespan extension and sufficient to extend lifespan regardless of diet.

### Traffic jam promotes *mtd* expression diet- and allele-specifically

In the DGRP, strains with the long-lived allele had elevated *mtd* expression in the head under DR, unlike strains that harbor the short-lived allele (Fig. 1j). Strains with the long-lived allele showed elevated expression of *mtd* transcripts that include exons 3' of the variant only (Supplementary Fig. 1i). However, transcripts with exons from both sides of the variant ("long transcripts") showed elevated expression on DR regardless of allele (Supplementary Fig. 1j). To validate the effects of these alleles, we cloned the allelic variant region into a LacZ reporter plasmid and transfected *w*[1118] wild-type flies (Fig. 1k). Upon rearing under DR, flies with the long-lived allele showed elevated LacZ expression versus flies reared under AL and versus flies transfected with the short-lived allele (Fig. 1l, m), suggesting that this genetic sequence influences nutrient-dependent changes in *mtd* expression. Using publicly available fly ChIP-seq data[12,13], we found a significant binding signal at this locus for CCCTC-binding factor (CTCF) and Traffic jam (TJ) (Supplementary Fig. 1k). Inducible neuronal *ctcf*[RNAi] increased *mtd* expression (Supplementary Fig. 1l), whereas *tj*[RNAi] reduced expression of all *mtd* transcripts under DR (Fig. 1n) and shortened lifespan specifically under DR (Fig. 1o). We validated that TJ, the fly homolog of MAF, binds this locus via ChIP-PCR (Supplementary Fig. 1m). In summary, we found that natural variants in mtd that show increased expression upon DR are regulated by TJ and are associated with lifespan extension upon DR, whereas CTCF represses *mtd* expression specifically under AL conditions, but that these effects only impact the shorter transcripts that are downstream of the variant site and TJ/CTCF binding site.

### *mtd* regulates the retromer complex

To define the pathway affected by loss and gain of Mtd/OXR1, we analyzed gene ontology (GO) terms for the top 50 genes that show co-expression patterns similar to human OXR1 (Supplementary Data 4)[14]. These showed significant enrichment for protein trafficking from the endosome to the lysosome or recycling endosomes (Fig. 2a and Supplementary Data 5). Neuronal *mtd*[RNAi] increased levels of Atg8a-II (fly ortholog of LC3), a marker for autophagosome formation (Supplementary Fig. 2a), consistent with the lysosomal accumulation phenotype observed in human fibroblasts and flies with *OXR1* deficiency[5]. We co-stained human fibroblasts[5] for OXR1 and markers for vesicular transport organelles (lysosome, endosome, Golgi, endoplasmic reticulum, and mitochondria). OXR1 staining overlapped most with the endosomal marker RAB7 (Supplementary Fig. 2b). As retromer dysfunction at the endosome causes lysosome dysfunction[15] as well as increased autophagosome formation[16] due to improper trafficking of endocytosed proteins and lipids, we hypothesized that OXR1 maintains retromer function. Neuronal *mtd*[RNAi] and *tj*[RNAi] significantly reduced the levels of retromer proteins (Fig. 2b and Supplementary Fig. 2c). This was not due to reduced retromer gene transcription under *mtd*[RNAi] (Supplementary Fig. 2d). Co-staining fly brains with antibodies against Mtd and retromer protein Vps35 with neuronal *mtd*[RNAi] also showed that Vps35 was reduced with *mtd*[RNAi] (Fig. 2c). We also observed co-localization of Vps35 and Mtd in the brains of *w*[1118] control flies (Fig. 2d), particularly in the optic lobes where OXR1 and the retromer have previously been shown to be relevant in flies[5,17]. We further determined that OXR1 interacts with VPS26A, VPS26B and VPS35 via co-immunoprecipitation in human fibroblasts (Fig. 2e). Retromer maintenance is necessary for neuronal health[18,19], and the loss of retromer proteins induces neurodegenerative disease progression[17,20]. Our results indicate that Mtd is involved in the maintenance of retromer function.

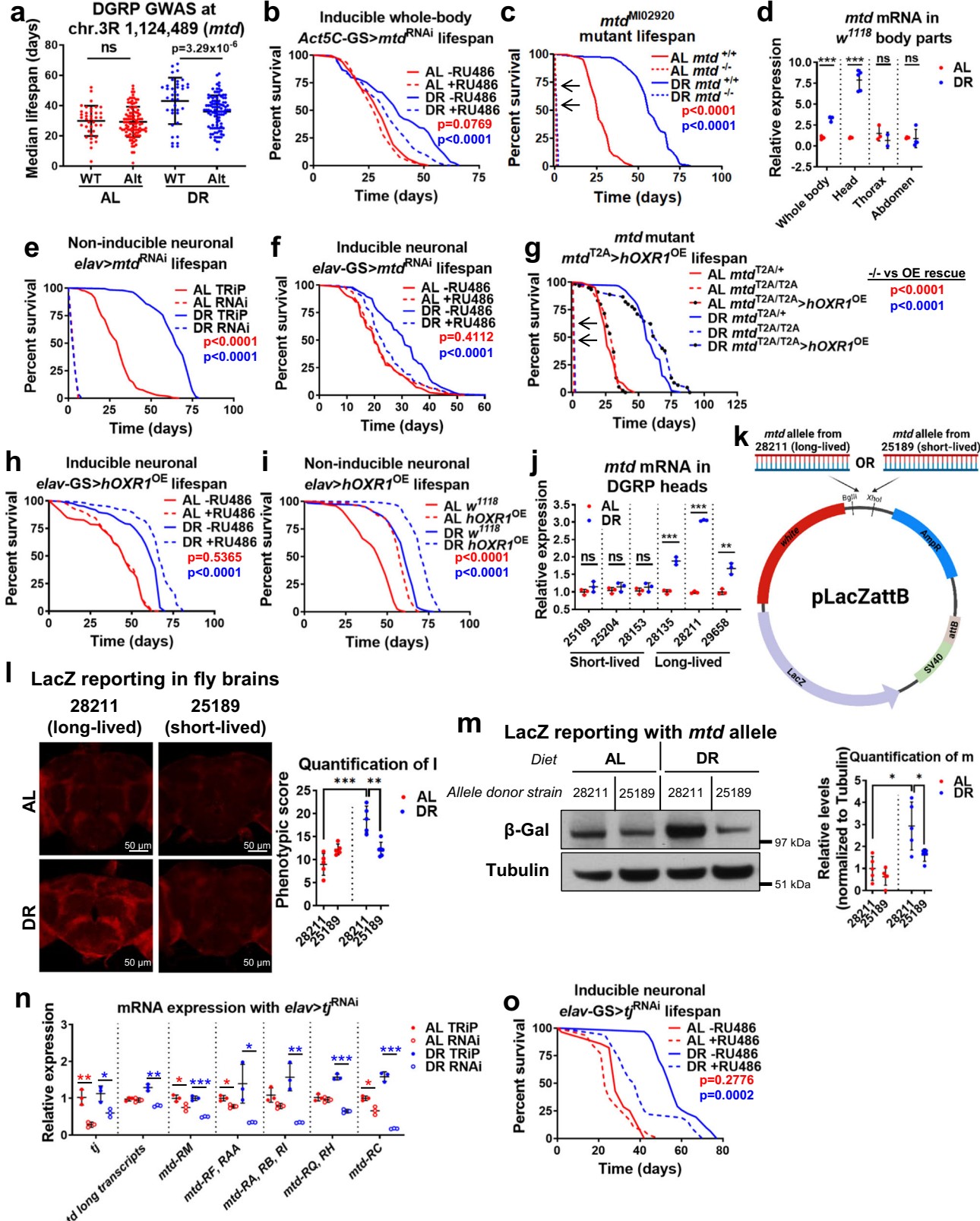

**Retromer stabilization rescues Mtd/OXR1 deficiency**

Next, we tested if enhancing retromer function rescues *mtd*^RNAi phenotypes. Neuronal overexpression (by *elav-Gal4* driver) of *ups26* or *ups35* rescued fly larval lethality (Fig. 3a) and lifespan (Fig. 3b, c, Supplementary Fig. 3a). Overexpression of *ups35* also recovered the loss of lifespan extension by DR in *mtd*^RNAi (Fig.3c). Moreover,

pharmacological retromer stabilization with the compound R55[21] also rescued developmental deformities (Fig. 3d) and the shortened lifespan of flies with *mtd*^RNAi (Fig. 3e). R55 further extended lifespan under DR beyond that of control strains on DR (Fig. 3e, f). However, simultaneous *mtd*^RNAi and overexpression of autophagy genes *Atg8*, *Atg6*, or *LAMP1* did not rescue developmental defects (Supplementary Fig. 3b),

**Fig. 1 | *mtd* is upregulated allele-specifically by Traffic Jam (TJ) to extend lifespan under DR. a** Alternate (Alt) *mtd* allele prevents lifespan extension by dietary restriction across DGRP strains. Dots represent median strain lifespan on AL (red) or DR (blue). Black bars represent mean across all strains. *n* = minimum 100 flies per strain. Data are presented as mean values across a single strain. Error bars represent mean value across all strains per condition +/- SD. **b** Conditional whole body *mtd*^RNAi in adulthood reduces lifespan under DR conditions. Dashed lines = RNAi induced by RU486, solid lines = ethanol vehicle control. *p* value determined by log-rank test. **c** Homozygous *mtd* null allele dramatically reduces lifespan. Dashed lines = *mtd*^MI02920 null allele strain, solid lines = *w^1118* control. p value determined by log-rank test. **d** *mtd* mRNA in *w^1118* fly head is upregulated by DR. Values normalized to AL. Samples taken after 7 days on AL or DR. *n* = 5 whole flies or abdomens, 50 heads, or 10 thoraces per biological replicate across minimum 3 independent experiments. Error bars represent mean value across replicates +/- SD. **e** Constitutively active pan-neuronal *mtd*^RNAi in development and adulthood dramatically reduces lifespan compared to TRiP (empty vector) control. *p* value determined by log-rank test. **f** Conditional pan-neuronal *mtd*^RNAi induced by RU486 in adulthood reduces lifespan only under DR conditions. *p* value determined by log-rank test. **g** Overexpression of human *OXR1*, driven by *mtd*^T2A-Gal4 rescues loss of *mtd*. Dashed lines = homozygous *mtd*^MI02920 null allele flies, solid lines = heterozygous controls, and dashed with circles = homozygous *mtd*^MI02920 null allele with *mtd-Gal4*-driven *hOXR1*^OE. *p* value determined by log-rank test. **h** Conditional pan-neuronal

overexpression of human *OXR1* induced by RU486 in adulthood extends lifespan under DR conditions. *p* value determined by log-rank test. **i** Constitutively active overexpression of human *OXR1* in development and adulthood extends lifespan. *p* value determined by log-rank test. **j**, DR increases *mtd* mRNA expression in heads of DGRP strains with the long-lived allele but not the short-lived allele. Samples taken after 7 days of AL or DR. *n* = 50 heads per replicate across 3 independent experiments. Error bars represent mean value across replicates +/- SD. **k** Schematic of LacZ reporter plasmid used for in vivo experiments in **l**–**m**. **l** LacZ staining is increased in whole brains from flies raised under DR transfected with cloned variant alleles from long-lived DGRP strains. Brains dissected after 7 days of AL or DR. *n* = 20 brains across 5 independent experiments. Error bars represent mean value across replicates +/- SD. **m** LacZ protein levels are increased in flies raised under DR transfected with long-lived variant allele reporter plasmid. Samples taken after 7 days of AL or DR. *n* = 20 heads per replicate across 5 independent experiments. Error bars represent mean value across replicates +/- SD. **n** Constitutively active pan-neuronal *tj*^RNAi reduces *mtd* expression in fly heads. Values normalized to AL. *n* = 20 heads per replicate across 3 independent experiments. Error bars represent mean value across replicates +/- SD. **o** Conditional pan-neuronal *tj*^RNAi induced in adulthood by RU486 reduces lifespan under DR. *p* value determined by log-rank test. For all figures, *$p < 0.05$, **$p < 0.005$, ***$p < 0.0005$. Except where noted, all *p* values were calculated by two-sided t-test. Figure 1k was generated using BioRender (publishing license: KW266MCH0G).

suggesting that defects due to *OXR1* deficiency are not due to a reduction in autophagic function.

Retromer stabilization by R55 rescued VPS35 levels and VPS35 co-localization with OXR1 in human fibroblasts from patients with *OXR1* mutations[5] (Fig. 3g). The mutation in *OXR1* also increased endosome size and number and impaired trafficking of mannose-6-phosphate receptor IGF2R from endosomes (EEA1 as marker) to the Golgi complex (GM130 as marker), all of which were rescued by treatment with 10 μM R55 (Fig. 3h). R55 also significantly reduced the number of lysosomes in the *OXR1*-deficient cells as well as localization of IGF2R with the lysosome (LAMP1 as marker) (Supplementary Fig. 3c, d). Lysosomal protease Cathepsin B was significantly increased in OXR1 disease fibroblasts and further increased with R55 (Supplementary Fig. 3e), despite the decreased lysosomal number in R55-treated cells. To further examine how lysosomal activity is impacted by OXR1-mediated retromer function, we compared LC3B-II levels in fibroblasts treated with the V-ATPase inhibitor bafilomycin, which blocks autophagy to demonstrate accumulated autophagosome pool size, to LC3B-II levels without bafilomycin, which represent steady-state pool size as LC3B-II is turned over during autolysosome formation[22]. *OXR1*-deficient fibroblasts had significantly reduced basal LC3B-II compared to control cells, whereas blocking LC3B turnover with bafilomycin showed significantly greater LC3B-II levels than control cells, indicating increased LC3B turnover in the *OXR1*-deficient cells. Treatment with 10 μM R55 increased the basal levels of LC3B-II in *OXR1*-deficient fibroblasts, rescuing the LC3B-II levels compared to the deficient cells not treated with R55 (Supplementary Fig. 3f, additional statistics in Supplementary Data 1). Neuronal RNAi for *vps26* and *vps29* in adulthood reduced lifespan (Supplementary Fig. 3g), with adulthood-specific *vps26*^RNAi reducing lifespan specifically under DR (Supplementary Fig. 3h). Together, these results suggest that OXR1 is necessary for retromer function, and retromer stabilization rescues altered autophagic activity in human cells with *OXR1* mutations and lifespan defects caused by *mtd*-deficiency in flies.

### Retromer stabilization rescues sensory decline and neurodegeneration induced by *mtd*^RNAi

To determine how *mtd/OXR1* deficiency influences brain aging, we analyzed the transcriptomes of heads from younger (day 7) and older (day 21) flies with and without neuronal *elav*-GS-*Gal4*-induced *mtd*^RNAi through principal component analysis (PCA). Transcriptomes from older *mtd*^RNAi DR flies grouped with older AL flies, suggesting that

*mtd*^RNAi accelerated transcriptional signatures towards older flies (Supplementary Fig. 4a and Supplementary Data 6, 7). GO term analysis for *mtd*^RNAi showed significant downregulation of genes associated with rhabdomere development and stimulus detection (Supplementary Fig. 4b and Supplementary Data 8), which have been implicated in retromer dysfunction in flies[17,23]. We observed delayed age-related visual decline upon DR, but this protective effect was abrogated by adulthood-specific neuronal *mtd*^RNAi (Fig. 4a). The ommatidia of the flies with constitutve RNAi were significantly more disordered than in TRiP empty vector controls (Fig. 4b). We assessed negative geotaxis but observed no change in activity between flies with or without neuronal *mtd*^RNAi, though activity was enhanced by DR (Fig. 4c), suggesting an uncoupling of activity and eye degeneration as well as lifespan. To determine if accelerated sensory decline was restricted to the visual system, we tested chemotaxis throughout lifespan. Neuronal *mtd*^RNAi under DR conditions induced more rapid decline in response to the attractant than DR controls, whereas flies under AL conditions were unaffected by RNAi (Supplementary Fig. 4c), suggesting impaired neuronal signaling for two phenotypes commonly disrupted with age, vision and olfaction[24,25]. Pan-neuronal *hOXR1*^OE improved phototaxis throughout life under DR (Fig. 4d). Retromer stabilization by *vps26*^OE or *vps35*^OE rescued phototaxis response in *mtd*^RNAi flies (Fig. 4e), as did treatment with 6 μM R55 (Fig. 4f). We observed no significant changes in body mass, triglyceride levels, or starvation resistance upon inducible pan-neuronal induction of *mtd*^RNAi (Supplementary Fig. 4d-f). TUNEL staining in whole fly brains showed that flies reared on DR have reduced cellular apoptosis with age (21 days), but that inducible adult-specific *mtd* RNAi negates this benefit of DR. This elevated degeneration by *mtd*^RNAi was rescued by overexpression of *vps26* or *vps35* in both diets (Fig. 4g, Supplementary Fig. 4g). These results indicate that *mtd* and the retromer are necessary for delaying age-associated degeneration in the fly brain and sensory systems and attraction to stimuli, and that *mtd* does not affect triglyceride metabolism.

### *mtd/OXR1* expression protects against neurodegenerative diseases and Alzheimer's disease phenotypes

To identify disorders associated with *mtd/OXR1*, we determined proteins significantly downregulated in flies with neuronal *mtd*^RNAi via data-independent acquisition mass spectrometry (q value < 0.05, Supplementary Data 9, 10)[26]. This revealed significant enrichment for proteins involved in multiple human neurodegenerative diseases

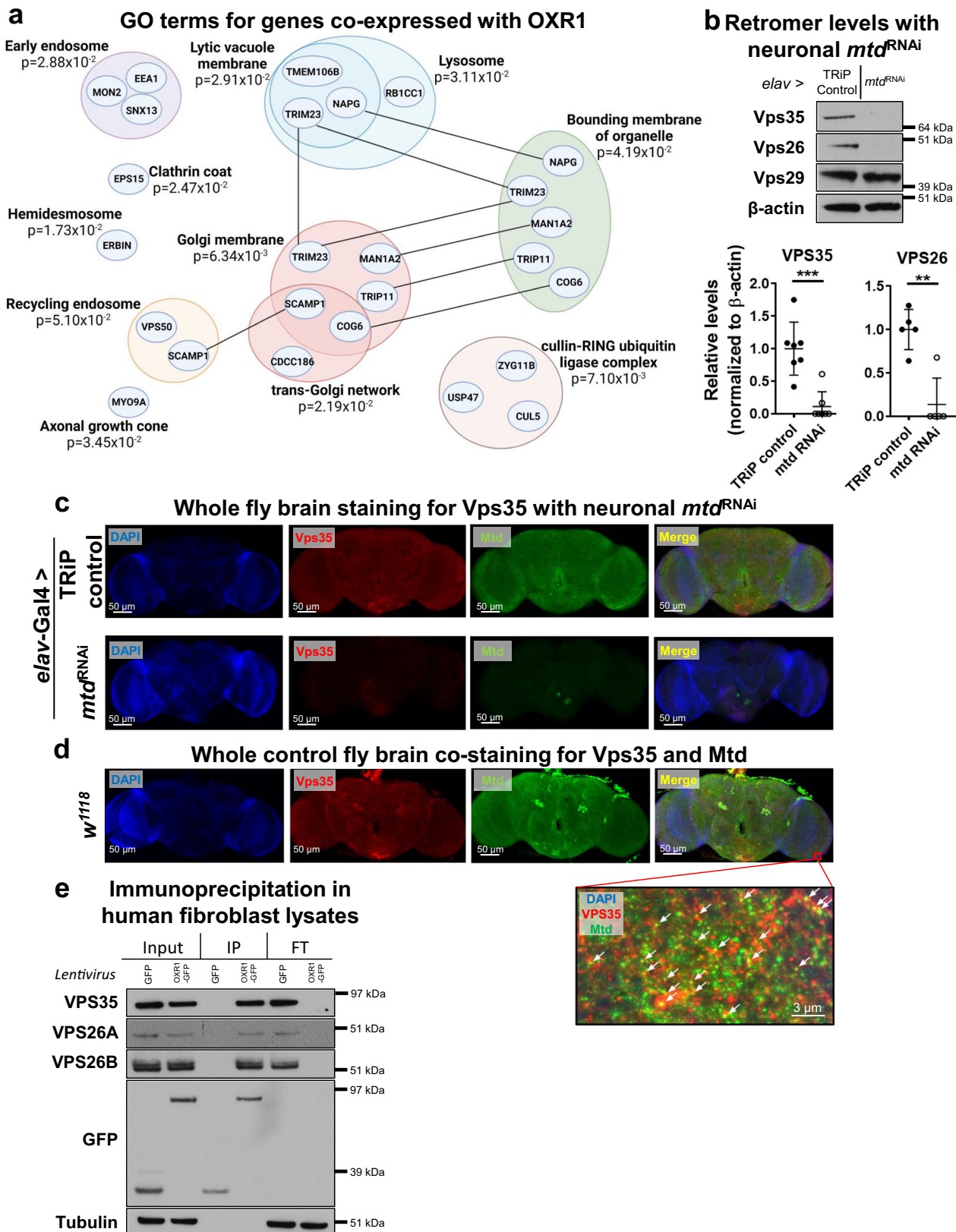

**a** GO terms for genes co-expressed with OXR1

**b** Retromer levels with neuronal *mtd*^RNAi

**c** Whole fly brain staining for Vps35 with neuronal *mtd*^RNAi

**d** Whole control fly brain co-staining for Vps35 and Mtd

**e** Immunoprecipitation in human fibroblast lysates

(Supplementary Data 11). We next analyzed -omics datasets from humans to learn more about the conserved nature of OXR1 in neurodegenerative diseases. We found an overlap of 83 genes between proteins with abundance levels that positively correlate with OXR1 ($\rho > 0.30$, Supplementary Data 12) from the Accelerating Medicines Partnership – Alzheimer's Disease (AMP-AD) dataset[27] and genes affected by age from the Genotype-Tissue Expression (GTEx) project (Fig. 4h and Supplementary Data 13)[28,29]. KEGG pathway analysis for the genes, which overlapped between these datasets revealed significant enrichment for age-related neurodegenerative diseases, including Alzheimer's, Huntington's, and Parkinson's diseases, (Fig. 4i and Supplementary Data 14)[30]. Retromer dysfunction has

**Fig. 2 | *mtd* interacts with the retromer complex and is required for retromer maintenance. a** GO terms for the top 50 genes co-expressed with OXR1 show enrichment for endolysosomal network. Arranged by organelle and function. **b**, Vps35 and Vps26 levels are reduced in in heads with constitutively active pan-neuronal *mtd*[RNAi]. Quantification below. *n* = 5 heads per replicate per condition across minimum 3 independent experiments. Error bars represent mean value across replicates +/- SD. **c** Immunohistochemistry of brains from flies under constitutively active pan-neuronal *mtd*[RNAi] (bottom) show reduced Vps35 signal. Blue = DAPI, red = Vps35, green = Mtd. **d** Immunohistochemistry of *w*[1118] fly brains show co-localization of Mtd and Vps35. Image below merge is magnified from boxed region in whole-brain images. **e** Co-immunoprecipitation in lysates from human fibroblasts transduced with GFP alone or OXR1-GFP show interaction between OXR1 and retromer proteins VPS35, VPS26A, and VPS26B. IP immunoprecipitated sample, FT sample flow-through. For all figures, *$p < 0.05$, **$p < 0.005$, ***$p < 0.0005$. All p values were calculated by two-sided t-test. Figure 2a was generated using BioRender (publishing license: YW266MCNQ5).

been associated with age-related neurodegenerative diseases that are protected by DR, including AD[31–35], especially in trafficking cargo involved in disease progression[21] and maintaining synaptic function[36]. We therefore also analyzed human OXR1 and retromer-associated protein abundance using the AMP-AD dataset[37]. Lower levels of OXR1, VPS35, SNX5, and RAB7A were all significantly associated with AD diagnosis and poor memory and pathology scores (Fig. 4j and Supplementary Data 15). To determine if OXR1 or retromer overexpression could rescue phenotypes associated with AD and tau pathogenesis, we neuronally overexpressed *hOXR1* or *vps35* in flies with simultaneous *GMR*-driven pathogenic human *Tau* in the eyes. Flies with *hTau*[OE] alone showed rapid eye degeneration with no changes in overall tau levels, but neuronal *hOXR1*[OE] or *vps35*[OE] rescued this degeneration phenotype (Fig. 4k and Supplementary Fig. 4h–j). These results demonstrate that the proteins in OXR1's network play a critical and conserved role in humans and flies for brain aging and neurodegeneration.

## Discussion

The widespread health and longevity benefits of DR are well-established, but the mechanisms by which DR mediates brain-specific functions are poorly understood. We previously showed in flies that DR elicits protective effects on lifespan and healthspan at least partially through distinct mechanisms[4]. While nutrient signaling pathways, such as TOR and ILS, are potential anti-aging pathways with roles in enhancing metabolism and burning fat[38], the variability in response to DR across different individuals and in different tissues suggests there are still many molecular mechanisms that remain unelucidated[2]. While screening for DR-responsive mechanisms, we found a role for *mtd*/OXR1 in regulating neuronal health with age and in response to DR. Here, we showed that *mtd*/OXR1 is an essential gene for healthy brain aging. We found that alleles of *mtd* that are upregulated by DR are regulated by the transcription factor TJ, and that this expression is necessary for DR-mediated lifespan extension. Specifically, our form of DR primarily restricts overall protein intake. Under AL conditions, CTCF is upregulated, inhibiting the expression of *mtd* in flies. Thus, protein restriction allows for increased *mtd* expression. As elevated nutrient-related components have been linked to dysregulation of the retromer[39,40], our work provides a genetic and molecular mechanism for this regulation. Mtd/OXR1 interacts with and maintains the retromer complex, and retromer stabilization rescues neurodegenerative and longevity defects induced by *mtd*/OXR1 deficiency. Our results show that *OXR1* expression is necessary for DR-mediated lifespan extension, and that neuronal overexpression of *OXR1* is sufficient for extending lifespan. While there is a large body of work demonstrating the benefits of DR, relatively few studies have detailed cellular mechanisms of DR on the brain. Our work presents a mechanism by which DR slows brain aging through the necessary action of OXR1 in maintaining the retromer.

Retromer function is necessary for neuron survival[18,41,42]. Thus, elucidating novel retromer pathway components is valuable for promoting brain health, especially with aging. Retromer dysfunction has been associated with age-related neurodegenerative diseases that are protected by DR, specifically Alzheimer's[31,43,44] and Parkinson's diseases [17,45–47], especially in trafficking cargo involved in

disease progression, such as APP[21] and β-secretase[48]. Mtd/OXR1 provides a linchpin for this interactive relationship between retromer regulation and DR-mediated neuroprotection with age. Our findings with OXR1's interaction with and regulation of retromer function resemble those of PLA2G6, which interacts directly with VPS26 and VPS35[17]. PLA2G6 maintains the retromer to improve ceramide trafficking[17]. It remains to be explored by which mechanisms the components of the retromer are lost when OXR1 is absent. Our work also shows that the effects of OXR1 on the retromer complex influence the lysosomal phenotypes in cells from patients with *OXR1* deficiencies, and this phenotype was rescued in patient cells supplemented with R55. These findings suggest that *OXR1* is a necessary component to gain the benefits of autophagy, especially under DR conditions, and that these benefits are mediated by OXR1's regulation of retromer function. An intriguing area of future research would be to determine if retromer proteins, such as VPS26 and VPS35, are degraded at an increased rate as a result of elevated autophagic function in instances of *OXR1* loss.

We further demonstrate that the *OXR1* gene network, conserved in flies and humans, is an important target for brain aging and neurodegenerative diseases. Loss of *mtd* accelerated aging, neurodegeneration, and visual decline in flies, but this was rescued by retromer stabilization. Additionally, overexpression of *OXR1* rescued age-related visual decline, neurodegeneration, and transgenically-induced tauopathy in a fly model. It remains to be observed which specific proteins or lipids are being mistrafficked, which result in neurodegeneration and reduction of lifespan with loss of *OXR1*. It is also not yet known if variants in genes that regulate *OXR1* might be associated with AD, as it seems that overall age-related decline in *OXR1* is more likely the cause for its associations with neurodegeneration. Our proteomics data support this idea, as *OXR1* declines significantly with age in AD patients. Our proteomics results in flies and humans also show enrichment for necessary components of synapses, and thus, future work could focus on understanding how changes in recycling influence neuronal function at the synapse. Additionally, *OXR1* has been implicated through GWAS in regulating ceramide levels[49]. Ceramide buildup has previously been demonstrated as a cause of lifespan reduction under retromer dysfunction[17], which could be an interesting area for future analysis. In all, our work implicates OXR1 and its network as a key regulators and potential therapeutic targets to slow aging and age-related neurodegenerative diseases across flies and humans.

## Methods
### Genome-wide association analysis
We used DGRP release 2 genotypes and FlyBase R5 coordinates for gene models. We used only homozygous positions and a minor allele frequency of ≥25% to ensure that the minor allele was represented by many observations at a given polymorphic locus[50]. We grouped the DGRP lines reared on a diet into a top 25 in-case group and a non-case group (>41 days median lifespan in DR conditions). The genotype distributions at each locus in these groups were compared via Fisher's exact test using the SciPy.stats module in Python[51]. The genotypes were arranged in the contingency table configuration [[#Case-RefAllele/#CaseAltAllele], [#NonCaseRefAllele/#NonCaseAltAllele]].

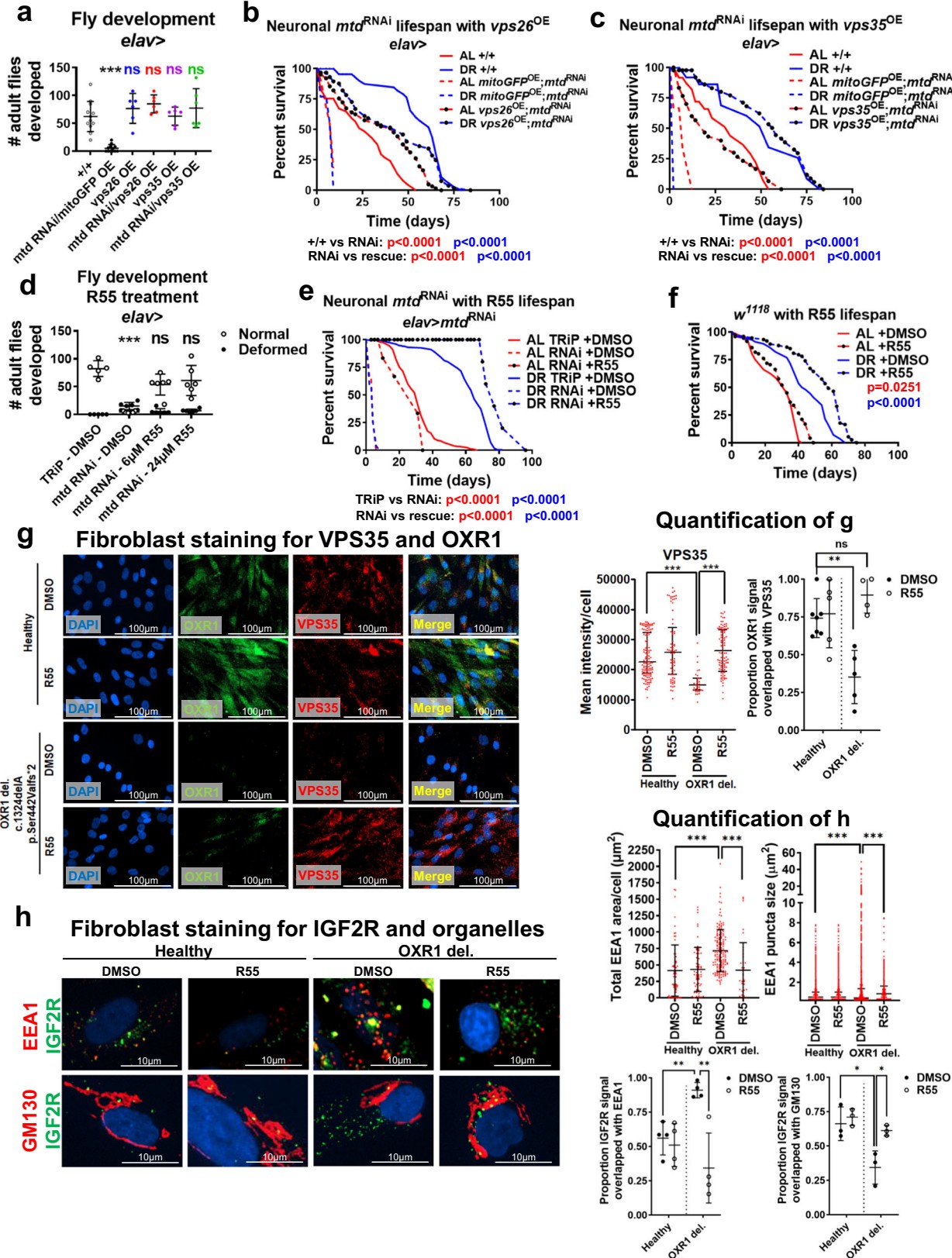

Nominal p-values denoted as "*P* value" in Supplementary Data 2 report the probability that the relative proportions of one variable are independent of the second variable. To avoid the potential for false positives at a given nominal cutoff owing to p-value inflation, we calculated false discovery rates via permutation as follows: for a given permutation i, we randomized phenotype values across DGRP lines, and assigned in- vs non-case designations on this permuted data set before we carried out Fisher's exact tests for each marker in turn as above. We counted the number of markers $n^i$ that scored above a given p-value threshold t. We tabulated the false discovery rate at t as the ratio between the average $n^i$ across 10 permutations and the number of markers called at t in the real data.

**Fig. 3 | Retromer stabilization rescues neuronal *mtd*/*OXR1* deficiency in flies and human fibroblasts. a** Retromer overexpression rescues developmental lethality induced by constitutively active neuronal *mtd*[RNAi]. *n* = at least 6 independent breeding chambers with 15 females to 5 males per chamber. Error bars represent mean value across replicates +/- SD. **b** Overexpression of *vps26* rescues lifespan defects in flies with constitutively active neuronal *mtd*[RNAi] to the levels of flies under AL conditions. p value determined by log-rank test. **c**, Overexpression of *vps35* rescues lifespan defects in flies with constitutively active neuronal *mtd*[RNAi] and restores DR-mediated lifespan extension. p value determined by log-rank test. **d**, R55 supplementation rescues developmental lethality induced by constitutively active neuronal *mtd*[RNAi]. *n* = at least 5 independent breeding chambers with 15 females to 5 males per chamber. Error bars represent mean value across replicates +/- SD. **e** 6 μM R55 supplementation throughout development and adulthood rescues lifespan defects in flies with constitutively active neuronal *mtd*[RNAi] and restores DR-mediated lifespan extension. p value determined by log-rank test. **f** 6 μM R55 supplementation throughout adulthood only extends lifespan. p value determined by log-rank test. **g** Human fibroblasts with a loss-of-function mutation in *OXR1* (c.132delA) have reduced VPS35 levels, which is rescued by 10 μM R55 supplementation. Blue = DAPI, green = OXR1, red = VPS35. Quantification on right. *n* = minimum 12 cells across minimum 4 independent experiments. Error bars represent mean value across replicates +/- SD. **h** Immunocytochemistry in human fibroblasts with loss-of-function *OXR1* mutation show enlarged endosomes (EEA1, red in top row), increased number of endosomes, and inability to traffic IGF2R to Golgi (GM130, red in bottom row). Quantification on right. *n* = minimum 38 cells across minimum 3 independent experiments. Error bars represent mean value across replicates +/- SD. For all figures, *$p < 0.05$, **$p < 0.005$, ***$p < 0.0005$. Except where noted, all p values were calculated by two-sided t-test.

## Gene expression analyses

To determine gene expression in a normal system, we sampled 50 heads from mated females of *w*[1118] control strain after 1 week on an AL or DR diet. We isolated RNA using Zymo Quick RNA MiniPrepkit (R1054) (Zymo Research, Irvine, CA). For qRT-PCR, we used Superscript III Platinum SYBR Green One-Step qRT-PCR kit from Invitrogen, Carlsbad, CA (11736-051) and followed the manufacturer's instructions with a Roche Lightcycler 480 II machine. Primers used are listed at the end of the Methods. To validate the effects of RNAi or mutation on gene expression, we collected 5 whole flies, 10 thoraces, 5 abdomens, or 50 heads after 1 week on AL or DR, depending on the body part being analyzed. We then isolated RNA from these samples and performed qRT-PCR on the perturbed genes as described above. RNA sequencing was performed by UC Berkeley QB3 Functional Genomics Laboratory, and reads were analyzed with the use of Bowtie2 read alignment package[52,53]. Two biological replicates per condition were used for RNA-seq analysis. Normalized reads counts are available in Supplementary Data Table S6.

## Fly strains used in this study

All flies were obtained from Bloomington Stock Center[54], Vienna *Drosophila* Resource Center[55], or FlyORF[56] and outcrossed six times to our lab control strains. Each line was mated and developed on a standard laboratory diet (1.5% yeast). At 2–3 days post-eclosion, unless otherwise noted, mated female progeny were transferred to AL (5.0% yeast extract) or DR (0.5% yeast extract) diet via $CO_2$[57]. Living flies were transferred to fresh vials every other day, and dead flies were recorded until all flies were deceased. Flies were maintained in a 12-h light/dark cycle in a room maintained at 25°C and 65% relative humidity[58]. A detailed list of fly strains used can be found at the end of the Methods section.

## Generation of LacZ transgenes

The 0.41 kb mtd gene region fragment was generated by genomic PCR cloning of BglII-Xho I fragment into the BglII-XhoI site of pLacZattB. The genomic DNA was isolated from the short-lived strain BL25189, and PCR was performed with pLacZ primers (listed below). The 0.41 kb genomic PCR was generated using the same primers with genomic DNA isolated from the long-lived BL28211 strain as template. Transgenic insertion was performed by Rainbow Transgenic Flies (Camarillo, CA). The transgenes were verified by sequencing with pLacZ primers (listed below). The primers used for cloning and sequencing are listed below.

## Fly phenotyping

We used the whole-body GeneSwitch driver *Act5C-GS-Gal4*, neuron-specific inducible driver *elav-GS-Gal4*, constitutively active *elav-Gal4* driver, glia specific *repo-Gal4*, or eye-specific *rdgA-Gal4* for directed RNAi. 15 virgin driver females were mated with three transgene line males in four bottles containing a standard diet. At 2–3 days after progeny eclosion, mated females were sorted onto AL or DR media with 200 μM RU486 (final concentration) for inducible RNAi activation[58,59] or ethanol as vehicle control, and flies were maintained on these media for life. Body mass was measured in five replicates of three flies using a Radwag 82/220.X2 analytical balance. Flies were then frozen, and triglycerides were measured using Stanbio Triglycerides LiquiColor Test. Glucose was measured using Stanbio Liqui-UV Hexokinase kit. For both assays, absorbance was measured using SpectroMax M2 spectrophotometer. Starvation was performed by rearing adult flies for 10 days on an AL diet and then transferring to medium containing water and 1% agarose[58]. Flies were checked for deaths every 4 h until all flies were deceased. Pupariation rate was determined by mating flies and placing 60 resulting embryos on an AL diet. Number of pupae formed was recorded daily. For western blot analysis, fly heads were homogenized in TPER lysis buffer and protein concentrations were determined by BCA. Western blots were quantified using ImageQuant. List of antibodies is provided at the end of Methods section. Negative geotaxis was performed by placing flies in empty vial and counting proportion of flies that climbed 6 cm within 10 sec. Three replicates were performed per vial. Phototaxis was performed by placing flies in on end of a 30-cm horizontal tube and shining light at one end. The proportion of flies that had travelled to the last 10 cm at the other end was recorded[4]. Chemotaxis was performed by placing flies in a tube with one end containing a cotton plug saturated with 1 mL of 1-hexanol attractant, the other end with water. The proportion of flies in the segment with 1-hexanol after 30 sec was recorded[60]. Starvation, lifespan, phototaxis, chemotaxis, and geotaxis assays were performed with approximately 200 starting adult flies, except for constitutively active *mtd*[RNAi] and *mtd* mutant flies, since few flies reach development in these strains. Rough eye phenotype was analyzed using Flynotyper plug-in in Image J[61]. Images were taken with Olympus BX51 with 10X objective using fiber optics gooseneck microscope illuminator. 10-12 optical slices were taken and reconstitute using Zerenestacker (Zerene Systems, Richland, WA).

## Cell lines used in this study

Patient cell lines were generated from skin biopsies and maintained in DMEM supplemented with 15% fetal bovine serum and 1% antibiotic cocktail[5]. Cells were maintained at 37°C. Cells undergoing serum withdrawal were treated with medium without fetal bovine serum. The OXR1 deletion line is from a female patient. Cell lines and media are described at the end of the Methods.

## Fibroblast lysates

Treatments with 10 μM R55 (Sigma-Aldrich #531084) were conducted for 48 h. Bafilomycin (100 nM, Cayman Chemical #11038) or DMSO control (Sigma Aldrich #276855) treatment was for 4 h prior to lysate collection. Lysates were collected with MPER lysis buffer and sonicated. Protein concentrations were determined by BCA. Western blots were quantified using ImageQuantTL.

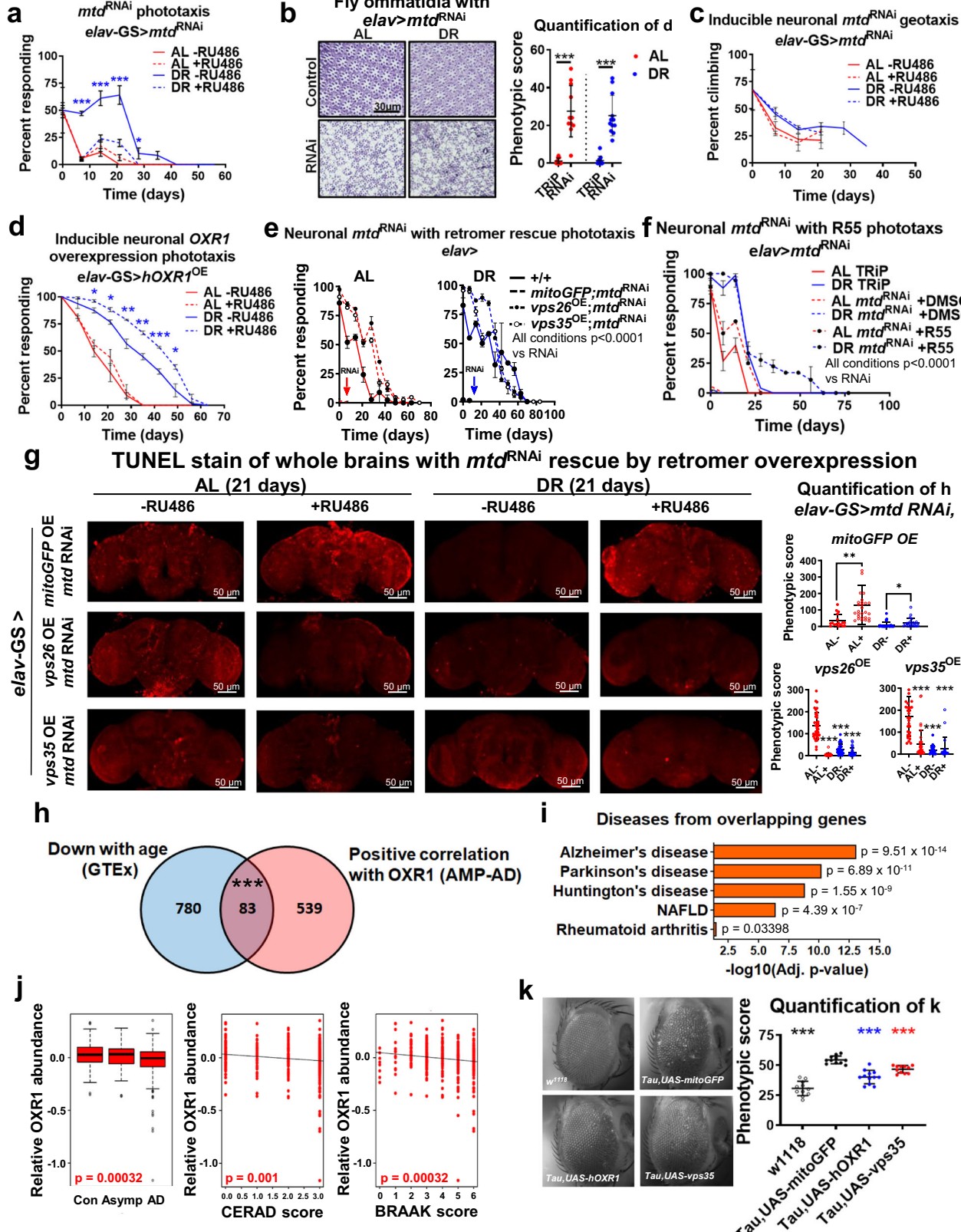

**g** TUNEL stain of whole brains with *mtd*^RNAi rescue by retromer overexpression

**i** Diseases from overlapping genes

| Disease | p-value |
|---|---|
| Alzheimer's disease | $p = 9.51 \times 10^{-14}$ |
| Parkinson's disease | $p = 6.89 \times 10^{-11}$ |
| Huntington's disease | $p = 1.55 \times 10^{-9}$ |
| NAFLD | $p = 4.39 \times 10^{-7}$ |
| Rheumatoid arthritis | $p = 0.03398$ |

## Co-immunoprecipitation

Fibroblasts were transduced with lentivirus overexpressing OXR1-GFP (Origene RC231430L4) or GFP alone (Origene PS100093V) for 2 days at 2 MOI. Lysates were collected, homogenized, and sonicated in MPER lysis buffer with protease inhibitors. Protein G beads (GE Healthcare/ Sigma 17061801) were washed in lysis buffer and samples were pre-

cleared with beads for 4 h at 4 °C to remove non-specific interactions. Sample-bead slurry was spun down, sample was added to fresh tubes, and 5 μL anti-GFP primary antibody (sc-9996) was added to sample to incubate on inverting rocker overnight at 4 °C. Pre-washed beads were added to the sample-antibody solution and incubated for 4 h at 4 °C on inverting rocker. The sample was spun down, and unbound protein

**Fig. 4 | Retromer stabilization by R55 or OXR1 overexpression preserves vision and rescues tauopathy-related dysfunction. a** Flies with RU486-induced neuronal *mtd*^RNAi in adulthood show reduced responsiveness to light. $n$ = minimum 50 flies per replicate across 4 independent experiments. Data represent mean value across replicates +/- SD. **b**, Flies with constitutively active pan-neuronal *mtd*^RNAi in development have disordered eye ommatidia. Image quantification on right. $n$ = eyes from minimum 11 fly heads per condition. Error bars represent mean value across replicates +/- SD. **c** Negative geotaxis is unaffected in flies with RU486-inducible neuronal *mtd*^RNAi. $n$ = at least 50 flies across 2 independent experiments. Data represent mean value across replicates +/- SD. **d** Flies with RU486-inducible pan-neuronal overexpression of human *OXR1* in adulthood have improved phototaxis throughout the lifespan. $n$ = at least 50 flies across 3 independent experiments. Data represent mean value across replicates +/- SD. **e** Overexpression of *vps26* or *vps35* rescues loss of phototaxis in flies with constitutively active neuronal *mtd*^RNAi. Dashed lines = *mtd*^RNAi, dashed with solid circles = *mtd*^RNAi with *vps26*^OE, dashed with open circles = *mtd*^RNAi with *vps35*^OE. AL shown on left (red) and DR shown on right (blue). $n$ = minimum 50 flies (except for *mtd*^RNAi due to larval lethality) per condition across 3 independent experiments. Data represent mean value across replicates +/- SD. **f** Supplementation with 6 µM R55 rescues loss of phototaxis in flies with constitutively active neuronal *mtd*^RNAi. $n$ = minimum 50 flies (except for *mtd*^RNAi due to larval lethality) per condition across 3 independent experiments. Data represent mean value across replicates +/- SD. **g** TUNEL stain for degeneration in aged fly brains is increased with RU486-inducible neuronal *mtd*^RNAi (top row) but rescued by overexpression of *vps26* (middle row) or *vps35* (bottom row). Flies were raised for 21 days under AL or DR with RU486 or vehicle control prior to sample collection. Quantification on right. $n$ = minimum 17 fly brains per condition. Error bars represent mean value across replicates +/- SD. **h** Venn diagram demonstrating number of genes downregulated in GTEx dataset and proteins positively correlated with OXR1 from AMP-AD dataset. Overlap in the middle. Significance determined by Fisher Exact test. **i** Alzheimer's disease, Parkinson's disease, and Huntington's disease are the most enriched diseases associated with the overlapping genes in **h**. $p$ value adjusted for multiple testing. **j**, Correlation of OXR1 protein abundance with AD diagnosis (left), CERAD score (middle), and BRAAK score (right). Asymp = asymptomatic AD. n = 453 human brains. Box plots represent the median, 25th, and 75th percentiles and whiskers represent the 5th and 95th percentiles. **k**, Constitutively active neuronal overexpression of human *OXR1* or *vps35* rescues degeneration induced by expression of mutant *Tau* driven in the fly eye with GMR promoter. Quantification of degenerative phenotype on right. $n$ = eyes from minimum 11 fly heads per condition. Error bars represent mean value across replicates +/- SD. For all figures, *$p < 0.05$, **$p < 0.005$, ***$p < 0.0005$. Except where noted, all $p$ values were calculated by two-sided t-test.

---

was collected as flowthrough sample (FT). Beads were washed 3 times with lysis buffer to ensure removal of unbound protein. The remaining sample and beads were incubated for 10 min at 95 °C with Invitrogen sample buffer and 1 M DTT for western blotting.

## LysoTracker

Fibroblasts were plated in 8-well chamber slides and maintained for 48 h before supplementation with R55 or DMSO control for another 48 h. LysoTracker Deep Red (Thermo Fisher Cat#L12492) was diluted 1:2000 in medium for lysosome staining and Hoechst 33342 solution (Thermo Fisher Cat#62249) was diluted 1:1000 in medium for nuclear staining. Cells were incubated with stains for 30 min at 37°C and then washed with medium and immediate imaged and analyzed using Bio-Tek Cytation 5 Imaging Reader. Mean intensity of red fluorescence per nucleus was quantified.

## Immunocytochemistry

After the treatments described above, cells were fixed in 4% paraformaldehyde in PBS. After three washes with PBS, cells were permeabilized using 0.1% Triton X-100 solution and blocked for 1 at room temperature with 1% bovine serum albumin and 5% donkey serum in PBS. Primary antibodies were diluted in blocking buffer, and cells were incubated overnight at 4°C. After three washes with PBS, secondary antibodies (1:1000) were added and incubated for 1.5 h at room temperature. Slides were mounted with ProLong Gold antifade reagent with DAPI (Invitrogen Cat#P36931). Cells were imaged and analyzed using BioTek Cytation 5 Imaging Reader. List of antibodies used is provided at the end of the Methods section.

## Immunohistochemistry

Adult *Drosophila* brains were dissected in PBS and immediately fixed in 4% PFA for 30 min. Post-fixation washes were done in PBS, and permeabilization was done in PT (PBS with 0.3% Triton X-100), three times for 30 min each. Blocking of tissues was done in permeabilization solution with 0.5% BSA. The brains tissue was labeled with primary antibodies (1:100) overnight in 4°C, followed by three washes with PT solution. Secondary antibody incubation was for 2 h at room temperature. Tissues were then washed three times with PT and mounted with Fluoromount G mounting medium. All incubation steps were done with continuous mild shaking. Images were taken in Zeiss LSM 780 confocal microscope. List of antibodies used is provided at the end of the Methods section.

## TUNEL staining

Adult *Drosophila* brains were dissected in PBS and immediately fixed in 4% PFA for 30 min. TUNEL staining was performed as per the manufacturer's instruction with few changes (Roche #11684795910). Post-fixation washes were performed in PBS and permeabilization was performed in 0.1% sodium citrate with 0.3% Triton X-100. Brains were incubated overnight with TUNEL solution and washed three times for 30 min each. Images were taken with Zeiss LSM 780 confocal microscope. Quantification was performed by counting TUNEL-positive cells per area.

## Proteomic analysis

Heads from adult flies with or without pan-neuronal *mtd*^RNAi were collected and flash frozen in liquid nitrogen in 5 replicates of at least 20 heads on the day of eclosion per replicate pool. Frozen heads from each condition/replicate were immersed in 100 µL of lysis buffer containing 4% SDS, 8 M urea, 200 mM triethylammonium bicarbonate (TEAB) at pH 8, 75 mM sodium chloride, 1 µM trichostatin A, 3 mM nicotinamide, and 1x protease/phosphatase inhibitor cocktail (Thermo Fisher Scientific, Waltham, MA). Lysates were subsequently homogenized for 2 cycles with a Bead Beater TissueLyser II (Qiagen, Germantown, MD) at 24 Hz for 3 min each, and further sonicated. Lysates were clarified by spinning at 15,700 x $g$ for 15 min at 4 °C, and the supernatants containing the soluble proteins were collected. Protein concentrations were determined using a Bicinchoninic Acid Protein (BCA) Assay (Thermo Fisher Scientific, Waltham, MA), and subsequently 100 µg of protein from each sample were aliquoted. Samples were then solubilized using 4% SDS, 50 mM TEAB at a pH 8. Proteins were reduced using 20 mM DTT in 50 mM TEAB for 10 min at 50 °C followed by 10 min at RT, and proteins were subsequently alkylated using 40 mM iodoacetamide in 50 mM TEAB for 30 min at room temperature in the dark. Samples were acidified with 12% phosphoric acid to obtain a final concentration of 1.2% phosphoric acid, and diluted with seven volumes of S-Trap buffer (90% methanol in 100 mM TEAB, pH ~7). Samples were then loaded onto the S-Trap mini spin columns (Protifi, Farmingdale, NY), and spun at 4000x $g$ for 10 s. The S-Trap columns were washed with S-Trap buffer twice at 4000x $g$ for 10 s each. A solution of sequencing grade trypsin (Promega, San Luis Obispo, CA) in 50 mM TEAB at a 1:25 (w/w) enzyme:protein ratio was then added, and after a 1-hour incubation at 47 °C, trypsin solution was added again at the same ratio, and proteins were digested overnight at 37 °C. Peptides were sequentially eluted with 50 mM TEAB (spinning for 1 min at 1000x $g$), 0.5% formic acid (FA) in water (spinning for 1 min

at 1000x $g$), and 50% acetonitrile (ACN) in 0.5% FA (spinning for 1 min at 4000x $g$). After vacuum drying, samples were resuspended in 0.2% FA in water and desalted with Oasis 10-mg Sorbent Cartridges (Waters, Milford, MA). All samples were vacuum dried and resuspended in 0.2% FA in water at a final concentration of 1 μg/μL. Finally, indexed retention time standard peptides, iRT (Biognosys, Schlieren, Switzerland)[62] were spiked into the samples according to manufacturer's instructions. The solvent system consisted of 2% ACN, 0.1% FA in water (solvent A) and 98% ACN, 0.1% FA in water (solvent B). Briefly, proteolytic peptides (2 μg) were loaded onto an Acclaim PepMap 100 C18 trap column with a size of 0.1 ×20 mm and 5 μm particle size (Thermo Fisher Scientific) for 5 min at 5 μL/min with 100% solvent A. Peptides were eluted on to an Acclaim PepMap 100 C18 analytical column sized as follows: 75 μm x 50 cm, 3 μm particle size (Thermo Fisher Scientific) at 0.3 μL/min using the following gradient of solvent B: 2% for 5 min, linear from 2% to 20% in 125 min, linear from 20% to 32% in 40 min, and up to 80% in 1 min, with a total gradient length of 210 min. Samples were analyzed by nanoLC-MS/MS in DIA mode[63,64] using a variable window isolation scheme[26] on the Orbitrap Eclipse Tribrid platform (Thermo Fisher Scientific, San Jose, CA). Samples were acquired in data-independent acquisition (DIA) mode. Full MS spectra were collected at 120,000 resolution (AGC target: 3e6 ions, maximum injection time: 60 ms, 350-1,650 m/z), and MS2 spectra at 30,000 resolution (AGC target: 3e6 ions, maximum injection time: Auto, NCE: 27, fixed first mass 200 m/z). The DIA precursor ion isolation scheme consisted of 26 variable windows covering the 350-1,650 m/z mass range with an overlap of 1 m/z[26]. DIA data were processed in Spectronaut v15 (version 15.1.210713.50606; Biognosys) using directDIA. Data were searched against the *Drosophila melanogaster* proteome with 42,789 protein entries (UniProtKB-TrEMBL), accessed on 12/07/2021. Trypsin/P was set as digestion enzyme and two missed cleavages were allowed. Cysteine carbamidomethylation was set as fixed modification, and methionine oxidation and protein N-terminus acetylation as variable modifications. Data extraction parameters were set as dynamic. Identification was performed using 1% precursor and protein q-value (experiment). Quantification was based on MS2 area, and local normalization was applied; iRT profiling was selected. Differential protein expression analysis was performed using a paired t-test, and p-values were corrected for multiple testing, specifically applying group wise testing corrections using the Storey method[65]. Protein groups with at least two unique peptides, $q$ value < 0.05, and absolute Log2(fold-change) > 0.58 were considered to be significantly changed comparing $mtd^{RNAi}$ to TRiP empty vector control strain, and are listed in Supplementary Data Table S10.

## Gene ontology analyses

GO term analysis of *Drosophila* genes was performed using the set of all genes downregulated on day 14 from our RNA sequencing dataset. Analysis was performed using Gene Ontology enRIchment anaLysis and visuaLizAtion tool (GOrilla) and Process, Function, and Component analyses were all represented. For genes co-expressed with human OXR1, the top 50 genes from ARCHS[4] tool[14] were analyzed using ENRICHR GO Cellular Component 2021 Ontology list[30].

## Statistics and reproducibility

Significance of differences between survival curves was assessed by log rank test. $p < 0.05$ was considered statistically significant. Error bars represent SD across at least three biological replicates. Significant differences between experimental groups and controls are indicated by *. *$p < 0.05$, **$p < 0.005$, ***$p < 0.0005$, determined by unpaired t test. nc = no change, ns = not significant. Significance for the Venn Diagram in Fig. 4h was calculated by Fisher's Exact Test. Statistical analyses were calculated with GraphPad Prism 4. Additional statistical details, including starting n for each experiment, can be found in Supplementary Data 1. Quantification of cell images was performed

using BioTek imaging software. Statistics for genome-wide analyses were performed as previously stated. For Fig. 2c, d, each condition was tested with 20 fly brains with similar results. For Fig. 2e, experiment was repeated 3 times with similar results.

| *Drosophila melanogaster* strains used | | |
| --- | --- | --- |
| **Strain** | **Source** | **ID number** |
| Drosophila Genetic Reference Panel strains | Bloomington Drosophila Stock Center | All DGRP strains |
| Act5C-GS-Gal4 Driver (inducible, whole body) P{w[+mC]=Act5C(FRT.y[+])GAL4.Switch.PR}X, y[1] w[*] | Bloomington Drosophila Stock Center | #9431 |
| Elav-GS-Gal4 Driver (inducible, neuronal) w1118; P{w[+mC]=elav-Switch.O} GSG301 | Bloomington Drosophila Stock Center | #43642 |
| Elav-Gal4 Driver (non-inducible, neuronal) P{w[+mW.hs]=GawB}elav[C155] | Bloomington Drosophila Stock Center | #458 |
| 5966-GS-Gal4 (inducible, intestinal) +; 5966-GS | Provided by lab of Dr. David Walker, University of California, Los Angeles | N/A |
| repo-Gal4 (non-inducible, glial) w[1118]; P{w[+m*]=GAL4}repo/ TM3, Sb[1] | Bloomington Drosophila Stock Center | #7415 |
| GMR-driven mutant Tau (non-inducible, eye), Elav-Gal4 (non-inducible, neuronal) P{w[+mW.hs]=GawB}elav[C155]; P{w[+mC]=GMR-htau/Ex]1.1 | Bloomington Drosophila Stock Center | #51360 |
| $w^{1118}$ control strain | Bloomington Drosophila Stock Center | #5905 |
| Transgenic RNAi Project (TRiP) empty vector control strain y[1] sc[*] v[1]; P{y[+t7.7] v[+t1.8] = VALIUM20-mCherry}attP2 | Bloomington Drosophila Stock Center | #35785 |
| mtd RNAi y[1] sc[*] v[1]; P{TRiP.HMS01666} attP2/TM3, Sb[1] | Bloomington Drosophila Stock Center | #38519 |
| mtd mutant y[1] w[*]; Mi{y[+mDint2]=MIC} mtd[MI02920]/TM3, Sb[1] Ser[1] | Bloomington Drosophila Stock Center | #76158 |
| hOXR1 overexpression w1118; P{UAS-OXR1.HA}1 | Bloomington Drosophila Stock Center | #64104 |
| hOXR1 overexpression w1118; P{UAS-OXR1.HA}4/TM3, Sb1 | Bloomington Drosophila Stock Center | #64105 |
| Fdxh RNAi w1118; P{GD1274}v24497 | Vienna Drosophila Resource Center | #24497 |
| CG15515 RNAi w1118; P{GD8577}v39872 | Vienna Drosophila Resource Center | #39872 |
| tj RNAi y[1] sc[*] v[1]; P{TRiP.HMS01069}attP2 | Bloomington Drosophila Stock Center | #34595 |
| TJ-GFP w{1118}; P33657Bac{y[+mDint2] w[+mC]=tj-GFP.FPTB}VK00033 | Bloomington Drosophila Stock Center | #66391 |
| ctcf RNAi y[1] sc[*] v[1]; P{TRiP.GL00266}attP2 | Bloomington Drosophila Stock Center | #35354 |
| ctcf RNAi y[1] v[1]; P{TRiP.HMS02017}attP40 | Bloomington Drosophila Stock Center | #40850 |
| vps26 RNAi y[1] v[1]; P{TRiP.HMS01769}attP40 | Bloomington Drosophila Stock Center | #38937 |
| vps29 RNAi y[1] v[1]; P{TRiP.HMJ21316}attP40 | Bloomington Drosophila Stock Center | #53951 |
| vps35 RNAi y[1] sc[*] v[1] sev2[1]; P{TRiP.HMS01858}attP40 | Bloomington Drosophila Stock Center | #38944 |
| Atg8 overexpression y[1] w[1118]; P{w[+mC]=UASp-GFP-mCherry-Atg8a}2 | Bloomington Drosophila Stock Center | #37749 |
| hLAMP1 overexpression y[1] w[*]; PBac{y[+mDint2] w[+mC] =UAS-hLAMP1.HA}VK00033 | Bloomington Drosophila Stock Center | #86301 |

| Atg1 overexpression y[1] w[*]; P{w[+mC]=UAS-Atg1.S}6 A | Bloomington Drosophila Stock Center | #51654 |
|---|---|---|
| mitoGFP overexpression w[1118]; P{w[+mC]=UAS-mito-HA-GFP.AP}2/CyO | Bloomington Drosophila Stock Center | #8442 |

| Cell lines and lentiviral vectors used | | |
|---|---|---|
| **Line** | **Source** | **ID** |
| F0062.1 Male human skin fibroblasts | | |
| Healthy | Provided by laboratory of Dr. Philippe Campeau, CHU Sainte Justine Research Center, Montreal, QC, Canada | N/A |
| F0342.1 Male human skin fibroblasts | | |
| OXR1 mutation | | |
| c.1324delA | | |
| p.Ser44Valfs*2 | Provided by laboratory of Dr. Philippe Campeau, CHU Sainte Justine Research Center, Montreal, QC, Canada | N/A |
| OXR1 (NM_001198533) Human Tagged ORF Clone | | |
| pLenti-C-mGFP-P2A-Puro vector | Origene | RC231430L4 |
| Lentiviral ORF control particles | | |
| pLenti-C-mGFP-P2A-Puro | Origene | PS100093V |

| Cell and fly media and additives used | | |
|---|---|---|
| Dulbecco's modification of Eagle's medium, Corning | VWR | Cat#45000-304 |
| Fetal bovine serum | Life Technologies | Cat#10082147 |
| Penicillin-streptomycin | Life Technologies | Cat#15140122 |
| Nutri-Fly Drosophila Agar, Gelidium | Genesee Scientific | Cat#66-104 |
| Yellow Cornmeal | Genesee Scientific | Cat#62-100 |
| Pure Cane Granulated | | |
| Sugar | C&H | N/A |
| Bacto yeast extract | VWR | Cat#90000-722 |
| Saf instant yeast | Rainy Day Foods | |

| Primers used | | |
|---|---|---|
| mtd Forward | Integrated DNA Technologies | GAAGAAGACTCCAAGGAGCT |
| mtd Reverse | Integrated DNA Technologies | CCCTATTTCTGCATCTAAGCG |
| Bgl2 mtd promoter Fr Forward | Integrated DNA Technologies | GGAAGATCTGTGTACATATTGAATCAAATCAGC |
| Xho1 mtd promoter Fr Reverse | Integrated DNA Technologies | TACGCTCGAGTTCTAGCCCTGTATCATACGG |
| pLacZ attB Forward | Integrated DNA Technologies | GAAGTTATGCTAGCGGATCC |
| pLacZ attB Reverse | Integrated DNA Technologies | GCGCCTCTATTTATACTCCGG |
| tj Forward | Integrated DNA Technologies | GATTCTGGTGAACACATCTTCGG |
| tj Reverse | Integrated DNA Technologies | TGGTGTGCGTAAGTCTGAGC |
| ctcf Forward | Integrated DNA Technologies | GAGCGCCAACTCCAAGATCA |
| ctcf Reverse | Integrated DNA Technologies | CCCATCGCCATACTCCTCAT |
| mtd gene region Forward (for TJ ChIP) | Integrated DNA Technologies | GACCTCGAAAGAGTCGCCAT |
| mtd gene region Forward (for TJ ChIP) | Integrated DNA Technologies | GATTCAGGGAATTGTGCGCC |
| mtd-RM Forward | Integrated DNA Technologies | TGGAAGACCTCGAAAGAGT |
| mtd-RM Reverse | Integrated DNA Technologies | CGAGTTCTCGGTTATCTACC |
| mtd long transcripts Forward | Integrated DNA Technologies | TCGACTTGGACTCGCTCCG |
| mtd long transcripts Reverse | Integrated DNA Technologies | TGGGTATGGTGGGCAATGAAG |
| mtd-RF, RAA transcripts Forward | Integrated DNA Technologies | GAGAGCCGGATAATCCACGA |
| mtd-RF, RAA transcripts Reverse | Integrated DNA Technologies | TCTCCATTGCGCCAGAAGAC |
| mtd-RA, RB, RI transcripts Forward | Integrated DNA Technologies | AAAAACTTACGGCCACGCTG |
| mtd-RA, RB, RI transcripts Reverse | Integrated DNA Technologies | TATTTCCCAGCGTCTCGTCG |
| mtd-RQ, RH transcripts Forward | Integrated DNA Technologies | GCTGAATATGTTCGCCGCC |
| mtd-RQ, RH transcripts Reverse | Integrated DNA Technologies | AAGCACTTGCAGAACATATAGAAAT |
| mtd-RC transcript Forward | Integrated DNA Technologies | GAAGAGCGGAAGGCGTAGAG |
| mtd-RC transcript Reverse | Integrated DNA Technologies | AAACGGCCAAGATGCCAAAC |
| hOXR1 Forward | Integrated DNA Technologies | GGTTTGCTGTGCCACAAG |
| hOXR1 Reverse | Integrated DNA Technologies | GGTTCTCTGGTATATTCGCCAG |
| RpL32 Forward | Integrated DNA Technologies | TAAGCTGTCGCACAAATGG |
| RpL32 Reverse | Integrated DNA Technologies | GGCATCAGATACTGTCCCT |
| Act5C Forward | Integrated DNA Technologies | AAGTACCCCATTGAGCACGG |
| Act5C Reverse | Integrated DNA Technologies | ACATACATGGCGGGTGTGTT |
| vps26 Forward | Integrated DNA Technologies | GGTATCCGGCAAGGTGAACG |
| vps26 Reverse | Integrated DNA Technologies | TTACCCCGGTCGTAGTACAGT |
| vps29 Forward | Integrated DNA Technologies | CGAGAACCTGACGTATCCGG |
| vps29 Reverse | Integrated DNA Technologies | AGGCCTCGAACTTGTACGTG |
| vps35 Forward | Integrated DNA Technologies | GCCATTGGACTAGCGAGGAAG |

| vps35 Reverse | Integrated DNA Technologies | AGCTCGTACAAATCCGTTTTCT | | |
| tau Forward | Integrated DNA Technologies | CCATCATAAACCAGGAGGTGGCC | | |
| tau Reverse | Integrated DNA Technologies | CTGTCTTGGCTTTGGCGTTCTC | | |

| **Antibodies and Stains** | | | | |
|---|---|---|---|---|
| Rabbit polyclonal anti-OXR1 | Invitrogen | Cat#PA5-72405 | WB concentration: 1:500 | Validated by RNAi |
| Rabbit polyclonal anti-OXR1 | abcam | Cat#ab103042 | ICC concentration: 1:100 | Validated against human OXR1 mutation |
| Goat polyclonal anti-VPS35 | abcam | Cat#ab10099 | WB concentration: 1:500 ICC concentration: 1:100 | Validated by KO (abcam) |
| Rabbit polyclonal anti-VPS26A | Proteintech | Cat#12804-1-AP | WB concentration: 1:500 | Previously validated[66] |
| Rabbit polyclonal anti-VPS26B | Proteintech | Cat#15915-1-AP | WB concentration: 1:500 | Validated by molecular weight |
| Rabbit polyclonal anti-VPS29 | Abcam | Cat#ab236796 | WB concentration: 1:500 | Validated by KO (abcam) |
| Mouse monoclonal anti-β-Galactosidase (LacZ) | Promega | Cat#Z3781 | WB concentration: 1:500 IHC concentration: 1:100 | Validated by molecular weight |
| Rabbit polyclonal anti-LC3B | Novus Biologicals | Cat#NB100-2220 | WB concentration: 1:150 | Validated by KO (Novus Biologicals) |
| Rabbit polyclonal anti-Atg8 | Sigma-Aldrich | Cat#ABC974 | WB concentration: 1:500 | Validated by molecular weight |
| Mouse monoclonal anti-Cathepsin B [CA10] | abcam | Cat#ab58802 | WB concentration: 1:500 | Validated by molecular weight |
| Rabbit polyclonal anti-GAPDH | abcam | Cat#ab9485 | WB concentration: 1:500 | Validated by molecular weight |
| Rabbit polyclonal anti-β-actin | Cell Signaling | Cat#4967 | WB concentration: 1:500 | Validated by company (Cell Signaling) |
| Mouse monocolonal anti-α-Tubulin [DM1A] | Sigma-Aldrich | Cat#T6199 | WB concentration: 1:500 | Validated by company (Sigma-Aldrich) |
| Mouse monoclonal anti-Rab7 [EPR7589] | abcam | Cat#ab50533 | WB concentration: 1:500 ICC concentration: 1:100 | Validated by company (abcam) |
| Mouse monoclonal anti-elav | DSHB | Cat#Elav-9F8A9 | IHC concentration: 1:100 | Validated by neuronal specificity |
| Mouse monoclonal anti-EEA1 [G-4] | Santa Cruz Biotechnology | Cat#sc-137130 | ICC concentration: 1:100 | Validated by company (Santa Cruz Biotechnology) |
| Mouse monoclonal anti-ATP5A [15H4C4] | abcam | Cat#ab14748 | ICC concentration: 1:100 | Validated by company (abcam) |
| Mouse monoclonal anti-LAMP1 [H4A3] | abcam | Cat#ab25630 | ICC concentration: 1:100 | Validated by company (abcam) |
| Mouse monoclonal anti-Calreticulin [FMC 75] | abcam | Cat#ab22683 | ICC concentration: 1:100 | Validated by company (abcam) |
| Mouse polyclonal anti-GM130 | abcam | Cat#ab169276 | ICC concentration: 1:100 | Validated by company (abcam) |
| Mouse monoclonal anti-GFP [B-2] | Santa Cruz Biotech | Cat#sc-9996 | WB concentration: 1:500 | Validated by molecular weight |
| Mouse monoclonal anti-Tau [HT7] | Invitrogen | Cat#MN1000 | WB concentration: 1:1000 | Validated by molecular weight |
| Sheep anti-Mouse IgG HRP-linked secondary | Sigma Aldrich | Cat#NXA931V | WB concentration: 1:3000 | |
| Donkey anti-Rabbit IgG HRP-linked secondary | Sigma Aldrich | Cat#NA934V | WB concentration: 1:3000 | |
| Mouse anti-Goat IgG HRP-conjugated secondary | Santa Cruz Biotech | Cat#sc-2354 | WB concentration: 1:3000 | |
| Alexa Fluor 488 donkey anti-rabbit IgG (H + L) secondary | Invitrogen | Cat#A21206 | ICC concentration: 1:500 | |
| Alexa Fluor 647 donkey anti-goat IgG (H + L) secondary | Invitrogen | Cat#A21447 | ICC concentration: 1:500 | |
| Alexa Fluor 647 donkey anti-mouse IgG (H + L) secondary | Invitrogen | Cat#A21238 | ICC concentration: 1:500 | |
| LysoTracker™ Deep Red | Thermo Fisher | Cat#L12492 | Live cell imaging: 1:2000 | |
| Hoescht 33342 Solution | Thermo Fisher | Cat#62249 | Live cell imaging: 1:1000 | |

## Reporting summary

Further information on research design is available in the Nature Portfolio Reporting Summary linked to this article.

## Data availability

The fly proteomics data generated in this study have been deposited in the MassIVE database under accession code MSV000088897 and ProteomeXchange ID: PXD031837 (Username: MSV000088897_reviewer; Password: winter) [https://massive.ucsd.edu/ProteoSAFe/dataset.jsp?task=d8f8a278868d42398c9ba8c772395c7b]. The processed fly proteomics and RNA sequencing data are provided in the Supplementary Information/Source Data file. Any additional information required to reanalyze the data reported in this paper is available from the lead contact upon request. Source data are provided with this paper.

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

## Acknowledgements

We would like to thank the members of the Kapahi and Ellerby labs as well as Dr. Malene Hansen, Dr. Gary Howard, and Ms. Kaitlyn Vitangcol for their comments throughout the manuscript and experiments. KAW was supported by NIH grant T32AG000266 and Buck CatalystX support from Bob and Alex Griswold. SB is supported by Larry L. Hillblom Foundation fellowship 2019-A-026-FEL. TAUH was supported by NIH grant 5F31AG062112. GC is supported by the DBT/Wellcome Trust India Alliance Fellowship/Grant IA/I(S)/17/1/503085. HJB is supported by NIH grants R01AG07326 and U01AG072439. Support of instrumentation for the Orbitrap Eclipse Tribrid from the NCRR shared instrumentation grant 1S10 OD028654, awarded to BS. LME is supported by NIH grant R01AG061879. PK is supported by NIH grant R01AG061165 and the Larry L. Hillblom Foundation. LME and PK are supported by NIH grant R56AG070705-01. We thank the UC Berkeley qb3 Functional Genomics Laboratory for the RNA sequencing referenced in Supplementary Fig. 1h, Supplementary Fig. 2d, Supplementary Fig. 4a, and Supplementary Fig. 4b. Figures 1k, 2a, Supplementary Fig. 1i, and Supplementary Fig. 4b were generated with the use of BioRender.com. BioRender licenses are listed in figure legends.

## Author contributions

Conceptualization: K.A.W., L.M.E., P.K. Methodology: K.A.W., S.B., E.B.D., G.C., R.B.B., H.J.B., B.S., N.T.S., L.M.E., P.K. Investigation: K.A.W., S.B., E.B.D., B.A.H., E.M.C., T.A.U.H., J.B., G.W.B., J.N.B., J.R., M.G.P., G.Q., J.L., C.S.N., A.A., G.C. Visualization: K.A.W., S.B., A.A.G., G.C. Funding acquisition: K.A.W., T.A.U.H., H.J.B., B.S., L.M.E., P.K. Project administration: P.M.C., L.M.E., P.K. Supervision: L.M.E., P.K. Writing – original draft: K.A.W., L.M.E., P.K. Writing – review & editing: K.A.W., S.B., E.B.D., B.A.H., E.M.C., T.A.U.H., J.B., G.W.B., J.N.B., J.R., M.G.P., C.S.N., G.Q., A.A., G.C., R.B.B., P.M.C., H.J.B., B.S., N.T.S., L.M.E., P.K.

## Competing interests

PK is founder and a member of the scientific advisory board at Juvify Bio. The other authors have no conflicts of interest.
