## [Peer Review File · Nature Communications]

OXR1 maintains the retromer to delay brain aging under dietary restrictionREVIEWER COMMENTS

Reviewer #1 (Remarks to the Author):

In this manuscript, the authors are investigating genetic mechanisms of Dietary Restriction (DR). They do a screen in flies and discover polymorphisms in the mustard gene, and pursued that it interacts with the retromere complex. They show that stabilizing the retromer prevents decreased lifespan in mustard flies, and that upregulation of retromere can protect against the rough eye of tau.

Overall these are interesting data and they have improved the manuscript based on previous reviews.

Some additional concerns:

Figure 1. in panel n, what is TrIP vs RNAi?

Figure 2, d is really hard to see.

In Figure 3, a-c, the number of UAS-transgenes needs to be balanced to rule out titration of GAL4 have odd effects.

In figure 3e, is the TrIP a control? Not clear.

In figure 4g, they should rule out a difference in the number of UAS-transgenes.

Reviewer #2 (Remarks to the Author):

The authors have addressed my latests comments. Including, performing additional studies, showing that in fibroblast OXR1 pulls down both VPS26 paralogs-- VPS26b and VPS26a. However, in the latest text, as far as I can tell, they do not include this addition info. For completeness sake, Please add this to text.

Reviewer #4 (Remarks to the Author):

I continue to stand by my previous reviews. There is a lot of very interesting data in this manuscript as well as (in my opinion) some remaining logical inconsistencies. However, the authors have responded in earnest to my comments, although, in certain instances, they see things a bit differently than me. I have no issue with that, and it seems at this point that it is reasonable to let the readers decide. I hope that the authors will consider addressing any outstanding comments, but I would also support publication without further revision.

Reviewer #1 (Remarks to the Author):

In this manuscript, the authors are investigating genetic mechanisms of Dietary Restriction (DR). They do a screen in flies and discover polymorphisms in the mustard gene, and pursued that it interacts with the retromere complex. They show that stabilizing the retromer prevents decreased lifespan in mustard flies, and that upregulation of retromere can protect against the rough eye of tau.

Overall these are interesting data and they have improved the manuscript based on previous reviews.

We thank the reviewer for their kind words, and are pleased that our efforts to improve the manuscript have been well-received.

Some additional concerns:

Reviewer 1 Point 1: Figure 1. in panel n, what is TrIP vs RNAi?

TRiP is the empty vector control strain that should be compared to the RNAi line. This has now been better stated in the text and figure legends. Thank you for this important clarifying question.

Reviewer 1 Point 2: Figure 2, d is really hard to see.

We apologize for this image being difficult to see. We have improved the clarity and size of this image.

Reviewer 1 Point 3: In Figure 3, a-c, the number of UAS-transgenes needs to be balanced to rule out titration of GAL4 have odd effects.

Thank you for this important point. We utilized a UAS-mitoGFP marker as a placeholder to ensure no change in Gal4 titration. The revised manuscript now states this more explicitly, and this strain has been added to the methods.

Reviewer 1 Point 4: In figure 3e, is the TrIP a control? Not clear.

Thank you for this point. As described above, TRiP is the empty vector control strain.

Reviewer 1 Point 5: In figure 4g, they should rule out a difference in the number of UAS-transgenes.

As described above, we have improved the figure to clarify that a UAS-mitoGFP line was used to account for this point. Thank you for taking the time to review our manuscript and make suggestions for its improvement.

Reviewer #2 (Remarks to the Author):

The authors have addressed my latests comments. Including, performing additional studies, showing that in fibroblast OXR1 pulls down both VPS26 paralogs-- VPS26b and VPS26a.

Reviewer 2 Point 1: However, in the latest text, as far as I can tell, they do not include this addition info. For completeness sake, Please add this to text.

This has now been added to the revised text. Thank you for your time and effort to improve our manuscript.

Reviewer #4 (Remarks to the Author):

I continue to stand by my previous reviews. There is a lot of very interesting data in this manuscript as well as (in my opinion) some remaining logical inconsistencies. However, the authors have responded in earnest to my comments, although, in certain instances, they see things a bit differently than me. I have no issue with that, and it seems at this point that it is reasonable to let the readers decide. I hope that the authors will consider addressing any outstanding comments, but I would also support publication without further revision.

We are pleased that the reviewer appreciates our responses, and that they acknowledge that it is reasonable for our readers to make their own interpretations. Thank you for your time and effort to improve the quality of our manuscript.